# Gut Microbiota’s Oxalate-Degrading Activity and Its Implications on Cardiovascular Health in Patients with Kidney Failure: A Pilot Prospective Study

**DOI:** 10.3390/medicina59122189

**Published:** 2023-12-17

**Authors:** Natalia Stepanova, Ganna Tolstanova, Iryna Aleksandrova, Lesya Korol, Taisa Dovbynchuk, Victoria Driianska, Svitlana Savchenko

**Affiliations:** 1State Institution “Institute of Nephrology of the National Academy of Medical Sciences of Ukraine”, 04050 Kyiv, Ukraine; lesyakorol@meta.ua (L.K.);; 2Educational and Scientific Institute of High Technologies, Taras Shevchenko National University, 01601 Kyiv, Ukraine; 3Educational and Scientific Centre “Institute of Biology and Medicine”, Taras Shevchenko National University, 01601 Kyiv, Ukrainetaisa.dovbynchuk@knu.ua (T.D.)

**Keywords:** oxalate-degrading activity, fecal microbiota, cardiovascular disease, kidney failure, kidney replacement therapy, plasma oxalate, indoxyl sulfate, lipid profile, oxidative stress, systemic inflammation

## Abstract

*Background and Objectives*: The present study aims to investigate the association between gut microbiota’s oxalate-degrading activity (ODA) and the risk of developing cardiovascular disease (CVD) over a three-year follow-up period in a cohort of patients undergoing kidney replacement therapy (KRT). Additionally, various factors were examined to gain insight into the potential mechanisms underlying the ODA–CVD link. *Materials and Methods*: A cohort of 32 KRT patients and 18 healthy volunteers was enrolled in this prospective observational pilot study. Total fecal ODA, routine clinical data, plasma oxalic acid (POx), serum indoxyl sulfate, lipid profile, oxidative stress, and proinflammatory markers were measured, and the patients were followed up for three years to assess CVD events. *Results*: The results revealed that patients with kidney failure exhibited significantly lower total fecal ODA levels compared to the healthy control group (*p* = 0.017), with a higher proportion showing negative ODA status (≤−1% per 0.01 g) (*p* = 0.01). Negative total fecal ODA status was associated with a significantly higher risk of CVD events during the three-year follow-up period (HR = 4.1, 95% CI 1.4–16.3, *p* = 0.003), even after adjusting for potential confounders. Negative total fecal ODA status was significantly associated with elevated POx and indoxyl sulfate levels and linked to dyslipidemia, increased oxidative stress, and inflammation, which are critical contributors to CVD. *Conclusions*: The findings contribute novel insights into the relationship between gut microbiota’s ODA and cardiovascular health in patients undergoing KRT, emphasizing the need for further research to elucidate underlying mechanisms and explore potential therapeutic implications of targeting gut microbiota’s ODA in this vulnerable population.

## 1. Introduction

Recent research has highlighted the significant role of gut microbiota in various aspects of human health, encompassing metabolism, immune function, and disease development [1,2]. Current studies have revealed that the composition and functionality of the gut microbiota can contribute to the pathogenesis of many chronic conditions, including chronic kidney disease (CKD) [3,4], cardiovascular disease (CVD) [5,6], and altered oxalate metabolism [7,8].

CVD represents a major global health concern, particularly among individuals with CKD [9]. Although traditional risk factors such as diabetes, hypertension, hyperlipidemia, and smoking have been extensively investigated, emerging evidence suggests a potential association between gut microbiota and CVD risk [5,6]. Specifically, oxalate-degrading bacteria (ODB), a subset of gut microbiota species, have garnered attention due to their role in oxalate metabolism, which is closely linked to atherosclerosis, CKD, and CVD progression [10,11,12]. However, the potential involvement of these bacteria in cardiovascular health remains largely unexplored.

The gut microbiota is involved in oxalate metabolism, with certain bacterial strains able to degrade oxalate and prevent its uptake in the gut. However, existing studies on ODB have primarily been conducted in vitro or animal models [13], leaving the relevance of these findings to human health uncertain. Among the known ODB are *Oxalobacter formigenes*, *Bifidobacterium*, *Enterococcus faecalis*, *Eubacterium lentum*, *Escherichia coli*, *Lactobacillus* spp., and *Providencia rettgeri* [8,13,14]. It is noteworthy that bacterial oxalate degradation involves a complex metabolic network comprising different taxa that interact with one another and the entire gut ecosystem, collectively influencing the capability of ODB to degrade oxalate [15,16]. This intricate interplay between bacterial species and their overall oxalate-degrading activity (ODA) within the gut microbiota is likely to play a significant role in intestinal oxalate handling and the development of hyperoxalemia [17]. Previous studies conducted by our team have shown that total fecal ODA, rather than the number of ODB alone, has a considerable impact on oxalate homeostasis in rats [17,18]. However, there is a general scarcity of clinical studies on this topic, with most of them focusing on the presence or quantification of ODB in coprofiltrate samples, which do not provide a comprehensive assessment of ODA. The only early study has directly measured the total ODA in fecal samples from patients with a jejunoileal bypass, utilizing a dilution solution containing [14C]-oxalate [19]. Therefore, a precise characterization of ODA in the human gut microbiota is still lacking and requires further investigation.

Kidney failure patients undergoing kidney replacement therapy (KRT) often experience hyperoxalemia, primarily as a result of the reduced renal excretion of oxalate [20,21,22]. In this context, gut microbiota may play a more significant role in maintaining oxalate balance compared to individuals with normal kidney function [23]. CKD is widely acknowledged to cause gut microbiota dysbiosis, which is evident in the early stages of CKD and progresses with structural and functional changes, culminating in distinct alterations during KRT [3,4,24,25]. Patients undergoing KRT exhibit a notable reduction in overall gut microbiota diversity, marked by a decline in anti-inflammatory butyrate-producing microbes (e.g., *Roseburia*, *Prevotella*, *Bacteroides*, *Eubacterium*) and an increase in pro-inflammatory microbes (*Proteobacteria*, *Actinobacteria*) [24,25]. These changes, coupled with commonly reported alterations such as decreased quantities of *Lactobacillaceae* and *Bifidobacteriaceae*, may impact the gut microbiota’s ODA and result in elevated plasma oxalic acid (POx) levels [7,23]. Concurrently, both gut microbiota dysbiosis and hyperoxalemia have been shown to be associated with CVD and all-cause mortality in patients with kidney failure [26,27,28]. CKD-induced gut microbiota dysbiosis exacerbates toxin accumulation, compromises the integrity of the mucosal barrier, disrupts immune tolerance, and intensifies systemic inflammation, contributing to the development of CVD and increased mortality in patients with kidney failure [24,29,30]. In turn, hyperoxalemia is hypothesized to potentially lead to endothelial dysfunction, inflammation, fibrosis, and altered hemodynamic parameters, suggesting a plausible pathophysiological link between oxalate and impaired cardiac function [28,31,32]. These findings prompt the exploration of whether gut microbiota ODA could serve as a potential link between hyperoxalemia and CVD in patients with kidney failure. However, despite significant evidence regarding the essential role of gut microbiota in both oxalate metabolism and CVD, the specific contribution of gut microbiota’s ODA to the development of CVD in this patient population has never been explored.

Therefore, the main objective of this study is to investigate the association between total fecal ODA and the risk of developing CVD over 3 years in patients undergoing KRT. Additionally, we aimed to explore the potential mechanisms underlying the ODA–CVD relationship.

## 2. Materials and Methods

### 2.1. Study Design

This prospective observational pilot study was conducted as one of the scientific projects undertaken by the state institution Institute of Nephrology of the National Academy of Medical Sciences (Institute of Nephrology) and Taras Shevchenko National University of Kyiv in Kyiv, Ukraine. The study adhered to the principles outlined in the Declaration of Helsinki and was carried out within the framework of the Institute’s research work on “The effect of oxalates and urates metabolism on the evolution of kidney disease” from January 2019 to December 2022 (ClinicalTrials.gov identifier NCT04399915, Domestic Trial Registration Identifier 0119U000002). The study protocol received approval from the local ethics committee at the Institute of Nephrology (Protocol #5 dated 12 June 2018). Informed consent was obtained from all participants before their inclusion in the study.

### 2.2. Sample Size

In this study, we encountered the limitation of not having previous data to conduct an a priori power analysis for determining sample size. Consequently, we took a pragmatic approach to establish the sample size, considering the recommended sample size of at least 50 participants for a pilot study [33] and practical factors related to the capacity of the commercially available POx and cytokine kits utilized in the study. Based on these considerations, we opted for a sample size of 50 measurements. By averaging 2 measurements per subject, we aimed to strike a balance between achieving a meaningful sample size and optimizing the utilization of the available resources.

### 2.3. Study Participants

The study involved a total of 50 participants. Among them, 32 were patients with kidney failure who had received regular dialysis treatment for at least 3 months, while the remaining 18 were healthy volunteers who served as the control group. Within the study group, 21 participants underwent hemodialysis (HD), and 11 received peritoneal dialysis (PD).

Inclusion criteria for the study group required patients to be 18 years of age or older, have a stable clinical condition, and possess a well-functioning arteriovenous fistula or peritoneal access. Additionally, the enrolled patients were expected to have a target Kt/V level of at least 1.4 for patients on HD and 1.7 for patients treated with PD. They should not have taken antibiotics or probiotics within the past three months. Patients with diabetes; those who had been hospitalized in the previous 3 months; or those who had a history of CVD events, peritonitis, anemia, systemic disease, malignancy, acute inflammation processes, or immunosuppressive treatment were not included in the study. These exclusion criteria were implemented to minimize the risk of other immune or inflammatory factors influencing the results of the total fecal ODA, oxidative stress markers, and cytokines under study.

All patients underwent their regularly prescribed dialysis treatment. The patients treated with HD received dialysis three times a week, with each session lasting 4 h. The HD procedure involved the use of bicarbonate-based dialysate, volumetric ultrafiltration control, and single-use synthetic (polysulfone) dialyzers. The minimal blood flow rate was 300 mL/min, and the dialysate flow rate was 500 mL/min. Heparin was administered as the standard anticoagulant. The patients treated with continuous ambulatory PD underwent a dwell time of 4–5 h during the day and 8–10 h at night. They were administered commercially available glucose-based Dianeal PD solution (Baxter Healthcare SA, Castlebar, Ireland) with varying glucose concentrations of 1.36% and 2.27%, along with Icodextrine.

### 2.4. Sample Collection

Blood samples were obtained from all participants after an overnight fast, simultaneously with the delivery of feces samples. For patients undergoing HD, blood samples were taken after the longest dialytic interval, whereas for patients treated with PD, blood samples were collected during the clinic visit. These blood samples were promptly processed to perform hematological and routine biochemical analyses, as well as to assess markers of lipid peroxidation and antioxidant protection.

To measure POx and cytokines, a 5 mL volume of blood was centrifuged at 2000 rpm for 15 min to separate the plasma/serum components. After centrifugation, the plasma and serum were carefully separated into 1.5 mL Eppendorf tubes for analysis. POx levels were analyzed immediately, while serum samples were stored at −20 °C until further cytokine analysis could be conducted.

Stool samples were collected by study participants in provided containers on the morning of sample submission and brought to our hospital for further processing within 4 h of defecation.

### 2.5. Clinical and Routine Laboratory Measurements

Demographic data including age, gender, comorbid conditions, and medication use were obtained from the medical records of the patients. Routine biochemical parameters, including blood concentrations of urea and creatinine, serum albumin, C-reactive protein (CRP), glucose, and lipid profile parameters, were measured using an automatic analyzer “Flexor Junior” (Vital Scientific, Spankeren, The Netherlands). Hematological parameters were measured using an “ABX Micros-60” (Horiba Medical, Montpellier, France). Parathyroid hormone (PTH) was measured using an immunoradiometric assay; electrolytes were measured using standard autoanalyzer techniques.

Blood lipid profile parameters consisted of triglyceride (TG), total cholesterol (TC), high-density lipoprotein cholesterol (HDL-C), and low-density lipoprotein cholesterol (LDL-C). The atherogenic index of plasma (AIP) was calculated from plasma triglyceride (TG) and HDL-C (log [TG/HDL-C]).

Body mass index (BMI) was calculated as weight in kilograms divided by the square of the height in meters.

### 2.6. Determination of ODA in Feces

Upon arrival at the facility, 0.01 g of feces was cultured in highly selective media called Oxalate Medium, consisting of the following components (g/L): K_2_HPO_4_—0.25, KH_2_PO_4_—0.25, (NH_4_)_2_SO_4_—0.5, MgSO_4_·7H_2_O—0.025, CH_3_COON—0.82, yeast extract—1.0, rezazurin—0.001, Na_2_CO_3_—4, L-cystein-HCl—0.5, and Trace element solution SL-10—1 mL (mix/L: HCl (25%; 7.7 M)—10.00 mL, FeCl_2_ × 4H_2_O—1.50 g, ZnCl_2_—70.00 mg, MnCl_2_ × 4H_2_O—100.00 mg, H_3_BO_3_—6.00 mg, CoCl_2_ × 6H_2_O—190.00 mg, CuCl_2_ × 2H_2_O—2.00 mg, NiCl_2_ × 6H_2_O—24.00 mg, Na_2_MoO_4_ × 2H_2_O—36.00 mg; Na_2_C_2_O_4_—5 mg) [34]. Then, the fecal samples were cultivated anaerobically at 37 °C for 48 h, creating a test solution.

To evaluate the total ODA in the fecal microbiota, the method of redoximetric titration with a KMnO_4_ solution was adapted [17]. Over 48 h, an aliquot of 10 mL of the test solution (or the Oxalate Medium as a control) was centrifuged, and the supernatant was transferred to a beaker. Calcium oxalate was precipitated by adding 10 mL of 0.4 M Ca(NO_3_)_2_. After filtration, the filtrate was discarded, and the precipitated calcium oxalate was dissolved in 25 mL of H_2_SO_4_ (1:4). The acidified calcium oxalate solution mixed with 20 mL of deionized water was heated to 80 °C before titration. Then, 10 mL of H_2_SO_4_ (1:4) solution was added, and the solution was titrated with KMnO_4_ (0.02 N) until a pink color persisted for 30 s. The results were expressed as the percentage of oxalate degradation per 0.01 g of feces.

### 2.7. Measurement of POx Concentration

POx concentration was measured spectrophotometrically using a commercially available kit (MAK315, Sigma, Madrid, Spain) according to the manufacturer’s protocols.

### 2.8. Determination of Plasma Malondialdehyde and Antioxidant Markers

Lipid peroxidation was assessed by measuring serum malondialdehyde (MDA) concentrations, expressed as µmol/L. To perform this measurement, 0.5 mL of serum was mixed with 1.5 mL of 0.025 M Tris buffer containing potassium chloride (pH 7.4) and incubated at 37 °C for 30 min. Subsequently, 1 mL of 17% trichloroacetic acid solution was added to the samples, which were then centrifuged at 3000× *g* for 20 min. Afterward, 1 mL of 0.8% thiobarbituric acid was added to the supernatant, which was then boiled at 100 °C for 10 min. After cooling, the absorbance of the samples was measured at 532 nm using a spectrophotometer. The 1,1,3,3-tetraethoxypropane was used as a standard to calculate the MDA concentration, expressed as µmol/L.

The antioxidant markers included serum ceruloplasmin, transferrin, and the number of sulfhydryl groups (SH groups). Serum ceruloplasmin was determined as follows: 0.05 mL of serum was added to 4 mL of 0.4 M acetic buffer solution (pH 5.5) and 0.5 mL of a 0.5% aqueous solution of 1,2-phenylenediamine dihydrochloride. The control sample (0.05 mL serum) was added to 1 mL of a 3% solution of sodium fluoride, 4 mL of acetate buffer, and 0.5 mL of a 0.5% aqueous solution of 1,2-phenylenediamine dihydrochloride. After incubation at 37 °C for 1 h, 1 mL of a 3% solution of sodium fluoride was added to the experimental sample. The absorbance was measured at 530 nm, and ceruloplasmin concentration was expressed in g/L.

Serum transferrin concentration was determined by adding 0.2 mL of serum to 2 mL of a 0.2% solution of ammonium-iron(III)-citrate (pH 5.5–5.8). The absorbance was measured at 440 nm after the first minute and after 30 min. Transferrin concentration was calculated as the difference between the absorbance readings. The result was expressed in g/L.

The level of SH groups in the serum was measured by dissolving 0.05 mL of serum in 0.5 mL of distilled water, adding 0.5 mL of 6 M potassium iodide solution, 2 drops of 5% starch solution, and 1.8 mL of 0.1 M phosphate buffer (pH 7.6). Absorbance was measured at 500 nm before and after the application of 0.3 mL of a 0.001 N iodine solution. The concentration of SH groups was expressed as mmol/L.

Based on the aforementioned markers, we also calculated the oxidative stress index (OSI) using the formula described previously [35].

### 2.9. Cytokine’s Measurements

Interleukin 6 (IL-6) and monocyte chemoattractant protein-1 (MCP-1) concentrations were detected in serum using “SunRise TouchScreen” enzyme and commercially available enzyme-linked immunosorbent assay (ELISA) test kits (IBL International GmbH, Hamburg, Germany). Cytokine analysis was carried out in accordance with the manufacturer’s protocol, with samples run in duplicate. The ELISA reader used for the measurements was the Tecan SunriseTM Absorbance Microplate Reader.

### 2.10. Total Serum Indoxyl Sulfate Determination

The concentration of total indoxyl sulfate (tIS) in blood serum was determined using a modified Obermeyer’s reagent method. Specifically, 0.5 mL of serum was mixed with 0.5 mL of 20% trichloroacetic acid solution and then centrifuged at 3000× *g* for 10 min. Next, a few drops of an alcoholic thymol solution and 1 mL of Obermayer’s reagent were added to 1 mL of the supernatant. After 20 min, 2 mL of chloroform was added, and the absorbance was measured spectrophotometrically at 450 nm. Indoxyl sulfate potassium was used as the standard for calibration, and the results were expressed in µmol/L.

### 2.11. Endpoint and Definition of CVD Events

Following baseline data collection and laboratory testing, the patients were monitored for 3 years to observe CVD events. For each patient, the time-to-event was calculated from the study’s entry date until the date of the first documented CVD event or the study’s completion date (31 December 2022).

CVD events were defined as newly diagnosed angina, myocardial infarction, stroke, heart failure, or peripheral artery diseases that necessitated hospitalization. The study’s flow is visually represented in Figure 1.

### 2.12. Statistical Analysis

The statistical analysis and graph generation were carried out using MedCalc Statistical Software version 22.007 (MedCalc Software Ltd., Ostend, Belgium). Descriptive statistics, such as average means (M), standard deviations (SD), or median (Me) with interquartile ranges (Q25–Q75), were calculated based on the variable’s distribution.

For data comparison, we used appropriate statistical tests depending on the distribution of the variables. Student’s *t*-test was employed for normally distributed data, and the nonparametric Mann–Whitney U test was used for non-normally distributed data. Categorical variables were presented as proportions, and the Chi-square test (χ^2^) was utilized for comparing the two groups.

To assess the association between total fecal ODA and other variables, we conducted Spearman’s correlation test.

The cut-off value for total fecal ODA to predict CVD events was determined via receiver operating characteristic (ROC) analysis.

To evaluate the association between total fecal ODA and the 3-year risk of CVD events, we performed univariate and multivariate Cox proportional hazard regression analyses. The univariate analysis included the patient’s age, sex, dialysis vintage, blood pressure, obesity, presence of anuria, and all studied markers that were significantly associated with ODA. Subsequently, the multivariate analysis was conducted to account for the confounding effects of the significant markers identified in the univariate analysis. We obtained Wald χ^2^, hazard ratios (HR), and 95% confidence intervals (CI) using the Cox proportional hazards regression models.

## 3. Results

### 3.1. Baseline Characteristics of the Study Participants

The participants of both groups were similar in age and sex and had similar BMI, total cholesterol, and glucose levels. However, other clinical, oxidative stress, and proinflammatory markers showed significant differences between patients with kidney failure and healthy volunteers, as expected. Along with the anticipated CKD-related outcomes, such as increased blood pressure, decreased GFR, anuria, electrolyte imbalances, anemia, and hyperuricemia, we observed notably elevated triglyceride levels and reduced levels of HDL-C, resulting in increased AIP when compared to the control group. Moreover, the study group exhibited statistically significant increases in MDA levels and decreases in antioxidant markers compared to the controls. Additionally, they showed elevated concentrations of tIS, POx, CRP, and cytokines when compared to the control group. The average duration of dialysis before enrollment was 38.5 (22–135) months. The most commonly prescribed medications included angiotensin-converting enzyme inhibitors or blockers of the renin–angiotensin–aldosterone system, erythropoietins, and iron supplementation, beta blockers, and statins. Non-calcium phosphate binders were the least frequently prescribed drugs among the participants of the study group. All data are presented in Table 1.

### 3.2. Microbiota’s ODA and Its Association with Examined Markers in the Entire Study Cohort

The total fecal ODA exhibited a wide range of values between −16% and 24% per 0.01 g of feces and was significantly lower in patients undergoing dialysis therapy compared to the healthy control group (Figure 2A). Negative total fecal ODA in the fecal microbiota, defined as ≤−1% per 0.01 g, was observed in 15 out of 32 (46.9%) individuals in the study group and 2 out of 18 (11.1%) individuals in the control group (χ^2^ = 5.5, *p* = 0.01) (Figure 2B).

In the entire study cohort, the ODA in the fecal microbiota showed a direct correlation with HDL-C levels and an inverse association with serum calcium levels, BMI, and triglycerides (Figure 3).

Furthermore, total fecal ODA was inversely associated with POx concentrations, tIS, serum MDA, and MCP-1, as shown in Figure 4.

Consequently, we also found an inverse correlation between ODA and OSI (r = −0.51, *p* = 0.002). Additionally, there was a direct association between total fecal ODA and serum ceruloplasmin levels (r = 0.45, *p* = 0.01).

### 3.3. Microbiota’s ODA and CVD Events in Patients with Kidney Failure

Over the 3-year follow-up period, 7 out of 32 (21.8%) patients experienced CVD events. Among them, 6 patients (85.7%) had non-fatal CVD events, while 1 patient died due to a stroke. Non-fatal CVD events included newly diagnosed angina (3 cases, 50%), heart failure (2 cases, 33.3%), and cardiac arrhythmia (1 case, 16.6%).

The ROC analysis showed that a negative total fecal ODA (≤−1%/0.01 g feces) was the most appropriate cut-off point for predicting CVD events in the patients treated with KRT, with a sensitivity of 77.8% and a specificity of 74% (Figure 5).

In the univariate Cox regression analysis, we found that negative total fecal ODA status, along with the patient’s age; dialysis vintage; obesity; and elevated levels of triglycerides, MDA, and proinflammatory cytokines (namely IL-6 and MCP-1, tIS, and Pox), as well as low levels of ceruloplasmin, were associated with CVD events in our study cohort (Table 2).

To assess the independent impact of these variables, we further conducted a multivariate model of Cox regression analysis, including all statistically significant variables obtained from the univariate regression. In this multivariate model, negative total fecal ODA status remained significantly associated with CVD events in patients undergoing KRT (Table 2, Figure 6).

### 3.4. Exploring the Link between Total Fecal ODA and CVD Events in Patients Undergoing KRT

To investigate the potential mechanisms underlying the relationship between total fecal ODA and CVD events, we analyzed POx and tIS concentrations, lipid profiles, oxidative stress, and proinflammatory markers according to the baseline status of total fecal ODA in the patients. As presented in Table 3, patients with negative total fecal ODA status had significantly elevated baseline triglyceride levels, lower HDL-C levels, and consequently, a higher AIP compared to the subgroup with positive ODA status.

Analysis of oxidative stress markers revealed higher levels of MDA and OSI, as well as decreased serum ceruloplasmin levels in patients with negative total fecal ODA status compared to the positive ODA status subgroup. Additionally, they exhibited higher concentrations of POx, tIS, CRP, and the studied cytokines (IL-6 and MCP-1) compared to patients with positive ODA status.

## 4. Discussion

In this prospective study, we aimed to investigate the association between total fecal ODA in patients undergoing KRT and the risk of developing CVD over 3 years of follow-up. Moreover, we examined various factors, including POx, tIS, lipid profile, oxidative stress, and proinflammatory markers, to gain insight into how microbiota’s ODA may impact cardiovascular health.

While attention toward gut ODB and their ODA has been growing, most of the existing data are from experimental studies or literature reviews [7,13,14,16,36,37]. To the best of our knowledge, our study represents the first clinical research shedding light on the potential role of gut microbiota’s ODA in influencing cardiovascular outcomes in patients with kidney failure undergoing KRT.

One noteworthy finding from our study was the significantly lower total fecal ODA and a higher proportion of patients with negative total fecal ODA status compared to healthy volunteers. The term “negative total fecal ODA status” of ≤−1% per 0.01 g was used to indicate that the gut microbiota’s ability to efficiently degrade oxalate in the fecal samples was severely impaired or non-existent. Essentially, these individuals had reduced capacity to break down oxalate in their gut, which can potentially lead to oxalate accumulation and hyperoxalemia.

The second main finding of our study demonstrated a significant association between the negative total fecal ODA status and the 3-year risk of CVD events in patients with kidney failure undergoing KRT. This association remained significant even after adjusting for multiple potential confounding factors. The link between gut microbiota’s ODA and CVD risk is intriguing and suggests a potential role of gut microbiota in influencing cardiovascular health in this vulnerable patient population. Although the exact mechanisms underlying this association remain to be fully elucidated, our study results propose several plausible explanations.

Consistent with previous research highlighting the crucial role of gut microbiota in oxalate homeostasis [14,15,19,37], we observed a negative association between total fecal ODA and POx concentrations, suggesting that ODA may influence POx concentration. As mentioned earlier, elevated levels of POx have been linked to atherosclerosis, oxidative stress, and inflammation [11,31,32,38], potentially impacting cardiovascular outcomes in patients with kidney failure. Moreover, hyperoxalemia has been shown to increase the risk of sudden cardiac death in patients treated with KRT [28], indirectly suggesting the possible ODA–CVD link.

The negative relationship between total fecal ODA and serum calcium concentration highlights the potential impact of gut microbiota on calcium metabolism and the development of hyperoxalemia. It has been demonstrated that gut microbiota may regulate calcium homeostasis and bone metabolism [39,40]. Consequently, gut bacteria with ODA can degrade oxalate, reducing its bioavailability and consequently minimizing POx levels whilst maintaining blood calcium balance [41]. In a reduction in ODA, there is a greater propensity for calcium to bind to oxalate, leading to a rise in serum calcium levels [41,42]. Elevated serum calcium in turn contributes to an increased risk of CVD and mortality in patients undergoing KRT [43].

Furthermore, our results revealed a significant inverse association between total fecal ODA and tIS levels. tIS is a gut-derived uremic toxin that accumulates in patients with kidney failure due to impaired renal clearance [44,45]. Elevated tIS levels have been associated with adverse cardiovascular outcomes, including increased risk of CVD and mortality in patients with kidney failure [46,47]. Our result suggests that gut microbiota’s ODA might play a role in tIS metabolism, potentially influencing its accumulation and contributing to cardiovascular risk. In addition, we observed that patients with negative total fecal ODA exhibited more severe dyslipidemia, characterized by significantly higher triglyceride levels and reduced HDL-C levels compared to those with positive ODA status. These lipid abnormalities are recognized cardiovascular risk factors, and their presence in patients with kidney failure could further contribute to cardiovascular complications [48,49]. The interplay between gut microbiota’s ODA, tIS, and blood lipid profile is complex and multifactorial [50,51]. One possible mechanism for the observed associations could be the gut microbiota’s ability to metabolize oxalate and indole compounds, affecting gut barrier integrity and leading to systemic inflammation [51,52]. Previous studies have shown that gut dysbiosis in patients with kidney failure can lead to increased gut permeability, allowing the translocation of microbial-derived toxins, including tIS, into the bloodstream [53,54]. This, in turn, can trigger systemic inflammation and oxidative stress, which might contribute to dyslipidemia and cardiovascular complications.

The association between total fecal ODA, oxidative stress markers, and proinflammatory cytokines represents another critical finding in our study, shedding light on the ODA–CVD relationship. Patients with negative total fecal ODA status demonstrated higher levels of MDA and OSI compared to those with positive ODA status, indicating increased lipid peroxidation and oxidative damage. Furthermore, they exhibited a decline in ceruloplasmin levels, which reflects compromised antioxidant defense systems. Ceruloplasmin is a copper-binding protein with potent antioxidant properties [55], and its reduction in patients with low ODA is in line with recently published studies that have linked altered gut microbiota to decreased ceruloplasmin and other antioxidant levels [56,57]. The observed inverse association between total fecal ODA and serum proinflammatory markers, such as CRP, IL-6, and MCP-1, suggests that gut microbiota’s ODA may exacerbate chronic inflammation, which is known to play a critical role in the progression of CVD in patients with kidney failure [58,59].

One possible avenue that emerges from our findings for therapeutic intervention is the use of prebiotics, probiotics, and synbiotics to modulate the composition and function of the gut microbiota, which could improve oxalate metabolism and reduce cardiovascular risk in this patient population. Several studies have demonstrated the use of pre-, pro-, and synbiotics in patients with kidney failure, showing promising results in modulating the composition of the gut microbiota, reducing uremic toxins and inflammation, and preserving residual renal function [29,60,61,62,63]. However, it is crucial to emphasize that the exact impact of interventions involving pre-, pro-, and synbiotics on oxalate metabolism [18,23] and the mitigation of cardiovascular risk [63] is not fully understood. This field of knowledge remains contentious and is continually evolving, necessitating further research to identify specific strains, dosages, and treatment durations [29,60,63].

The strengths of our study include its prospective design and 3-year follow-up period, which allowed us to assess the long-term association between total fecal ODA and CVD events. Additionally, we comprehensively explored various potential mechanisms underlying the ODA–CVD relationship by examining relevant clinical and biochemical factors.

However, as a pilot study, our research possesses several limitations that warrant acknowledgment. First, the small sample size and observational nature of our study design hinder the establishment of a causal relationship between total fecal ODA and cardiovascular outcomes and limit the generalizability of our findings to a broader population of patients with kidney failure undergoing KRT. Second, despite our efforts to adjust for potential confounding factors, there remains the possibility of unaccounted-for variables influencing the observed associations. Factors such as dietary habits, medication usage, and other lifestyle factors were not extensively evaluated in our study. Third, the lack of a standardized method for ODA measurement and the potential impact of interindividual variations in gut microbiota composition may introduce variability in ODA determination results. Fourth, while we explored the impact of total fecal ODA on cardiovascular outcomes, we did not analyze the specific composition of the gut microbiota. Understanding the role of individual bacterial species and their functional capacity in oxalate metabolism could offer more comprehensive mechanistic insights. Additionally, our study only assessed total fecal ODA at baseline, and we did not collect longitudinal data on ODA changes over time. Long-term follow-up data could provide more profound insights into the dynamic relationship between gut microbiota’s ODA and cardiovascular outcomes. Finally, the absence of detailed dietary information from our study participants limits our ability to assess the potential impact of oxalate intake and metabolism on cardiovascular outcomes. Further research incorporating dietary assessments may provide a more comprehensive understanding of the interplay between gut microbiota’s ODA, diet, and cardiovascular health.

Nonetheless, despite these limitations, our study provides valuable initial insights into the association between total fecal ODA and cardiovascular outcomes in patients with kidney failure undergoing KRT. Further research is warranted to overcome these limitations and build a more comprehensive understanding of the gut–kidney axis and its implications for cardiovascular health in this patient population.

## 5. Conclusions

Our prospective pilot study revealed a significant association between total fecal ODA and the risk of developing CVD over a 3-year follow-up period in kidney failure patients undergoing KRT. The associations between ODA and factors such as POx, serum calcium and tIS levels, dyslipidemia, oxidative stress, and inflammation suggest a complex interplay between gut microbiota and cardiovascular health. Further research is needed to elucidate the exact mechanisms underlying the ODA–CVD relationship and identify potential therapeutic targets to mitigate cardiovascular risk in this vulnerable patient population.

## Figures and Tables

**Figure 1 medicina-59-02189-f001:**
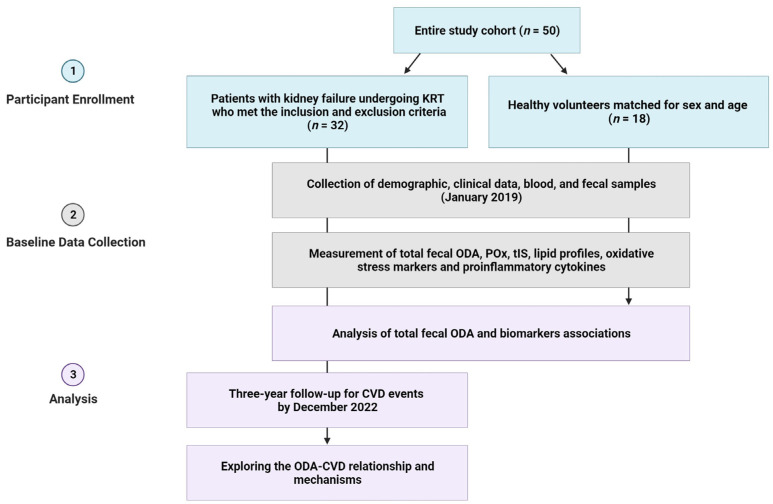
The study’s flow chart. Abbreviations: CVD, cardiovascular disease; KRT, kidney replacement therapy; POx, plasma oxalic acid; ODA, oxalate-degrading activity; tIS, total indoxyl sulfate.

**Figure 2 medicina-59-02189-f002:**
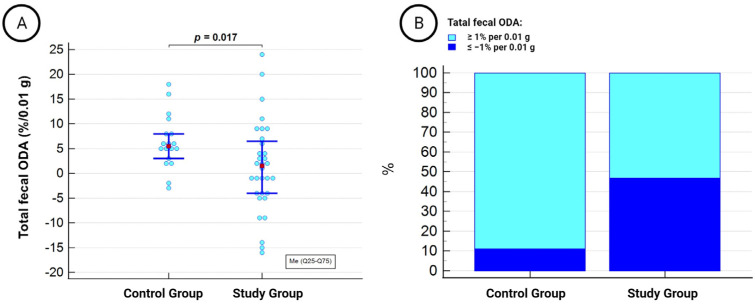
Median total fecal ODA (**A**) and its distribution according to positive and negative statuses (**B**) in patients with kidney failure compared to controls.

**Figure 3 medicina-59-02189-f003:**
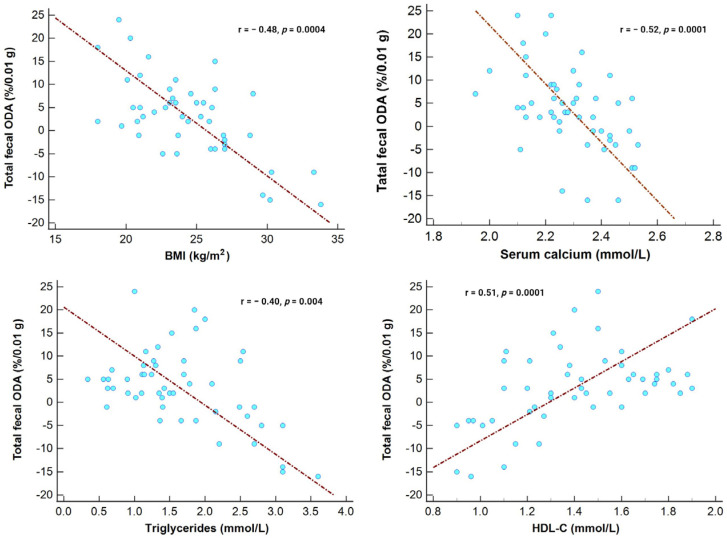
Correlation between total fecal ODA and HDL-C, calcium, BMI, and triglyceride levels in the entire study cohort.

**Figure 4 medicina-59-02189-f004:**
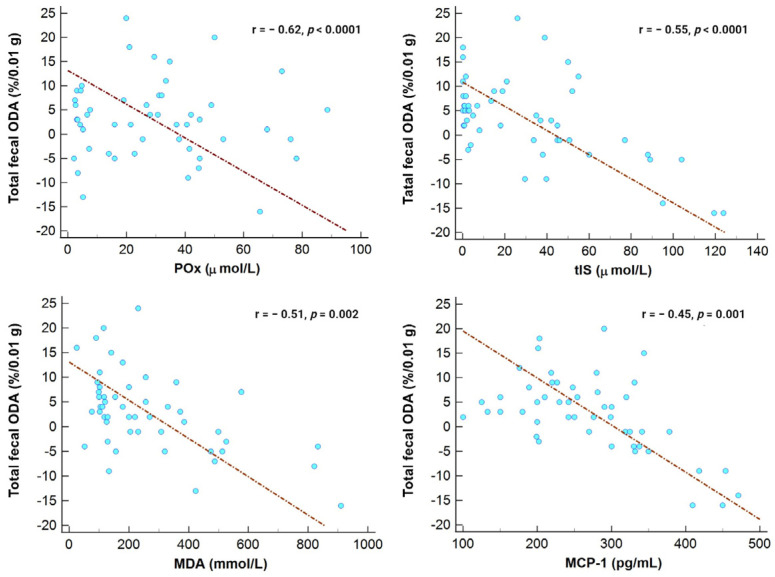
Correlation between total fecal ODA and POx, tIS, MDA, and MCP-1 levels in the entire study cohort.

**Figure 5 medicina-59-02189-f005:**
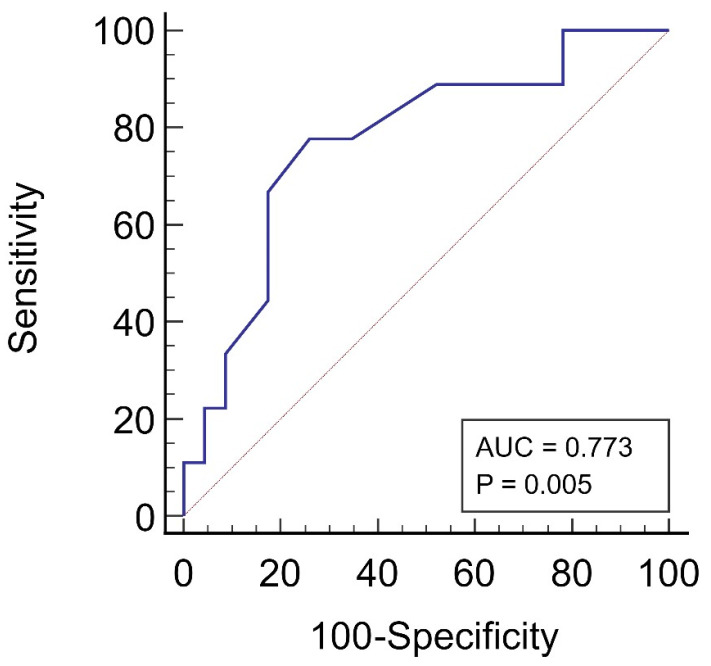
The ROC curve for the cut-off value of total fecal ODA for predicting CVD in patients with kidney failure undergoing KRT.

**Figure 6 medicina-59-02189-f006:**
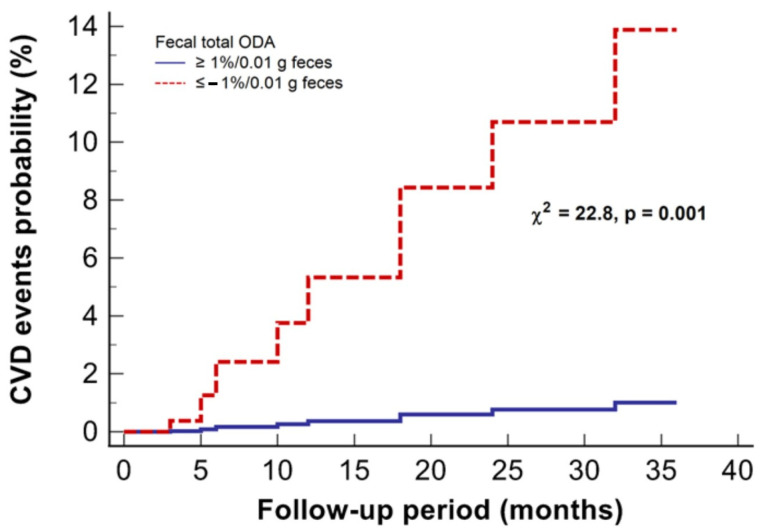
The 3-year probability curves of CVD events stratified by the status of total fecal ODA in patients with kidney failure treated with KRT.

**Table 1 medicina-59-02189-t001:** Baseline characteristics, oxidative stress, and proinflammatory markers of the study participants.

Clinical and Laboratory Data	Control Group (*n* = 18)	Study Group (*n* = 32)	*p*-Value
Demographic and clinical parameters			
Male sex, n (%)	6 (33.4%)	18 (56.2%)	0.12
Age, years	46.4 ± 10.9	49.1 ± 13.4	0.54
BMI, kg/m^2^	23.8 ± 4.2	24.4 ± 3.9	0.31
Systolic blood pressure, mm Hg	116.2 ± 10.9	128.9 ± 12.2	0.006
Diastolic blood pressure, mm Hg	73.2 ± 8.8	82.1 ± 10.3	0.003
Time on dialysis, months		38.5 (22–135)	
Anuria, n (%)		11 (34.4%)	
Serum creatinine (µmol/L)	87 (82.02–102.5)	751 (575.7–968.4)	<0.0001
Serum urea	5.4 (4.3–6.1)	22.5 (17.7–31.3)	<0.0001
GFR (mL/min/1.73 m^2^)	76.4 (66.1–79.8)	4.5 (3.5–7.0)	<0.0001
Serum albumin, g/L	42.2 (41.05–46.1)	40.6 (38.5–41.6)	0.003
Hb, g/L	123 (119.5–133)	105.5 (94–125)	0.003
Glucose, mmol/L	5.4 (4.7–5.7)	4.9 (4.02–5.9)	0.47
Calcium, mmol/L	2.37 ± 0.1	2.29 ± 0.18	0.019
Phosphorus, mmol/L	1.05 (0.9–1.1)	1.99 (1.5–2.6)	<0.0001
Potassium, mmol/L	4.1 (3.8–4.4)	5.9 (5.2–6.9)	0.0001
Uric acid (mmol/L)	286.7 (146.4–374)	329.4 (294.5–389.3)	0.007
Lipid profile parameters			
TC, mmol/L	4.9 (4.6–5.3)	5.3 (4.5–6.6)	0.15
Triglycerides, mmol/L	1.12 (0.9–1.6)	1.56 (1.1–2.5)	0.05
LDL-C, mmol/L	2.3 (1.5–2.9)	3.1 (2.5–3.5)	0.05
HDL-C, mmol/L	1.5 (1.3–1.65)	1.2 (1.07–1.56)	0.04
AIP	2.7 (1.9–3.1)	3.2 (2.5–4.9)	0.01
Oxidative stress markers			
MDA, mmol/L	119 (101.3–128.4)	256.4 (134.6–480.7)	<0.0001
Ceruloplasmin, g/L	0.22 (0.16–0.31)	0.13 (0.09–0.19)	0.003
Transferrin, g/L	5.2 (4.5–5.8)	2.2 (1.5–2.6)	<0.0001
SH groups, mmol/L	2.22 ± 1.1	1.65 ± 0.69	0.02
OSI, CU	1.04 ± 0.04	3.8 ± 2.1	<0.0001
Proinflammatory markers, tIS and POx concentrations			
CRP, mg/L	4.2 (2.9–10.9)	11.6 (7.7–17.8)	0.001
IL-6, pg/mL	0.8 (0.5–1.3)	2.5 (1.9–6.1)	0.0001
MCP-1, pg/mL	200.0 (159.7–239.1)	327.5 (281.2–344)	<0.0001
tIS, μmol/L	0.85 (0.5–2.6)	39.8 (20–51.3)	<0.0001
POx, µmol/L	4.5 (3.09–7.3)	39.3 (26.2–49.5)	<0.0001
Medications, n (%)			
ACE inhibitors/RAAS blockers		25 (78%)	
Beta blockers		18 (56.2%)	
Calcium channel blockers		7 (21.9%)	
Iron supplementation		19 (59.3%)	
Erythropoietins		23 (71.8%)	
Calcium-based phosphate binders		9 (28.1%)	
Non-calcium phosphate binders		3 (9.4%)	
Statins		11 (34.4%)	

The values are expressed as mean ± standard deviation (M ± SD) or as the median and interquartile range (Me (Q25–Q75)). The values are compared between the groups using the Chi-square test, Student’s *t*-test, and the Mann–Whitney U test as appropriate. Abbreviations: ACE, angiotensin-converting enzyme; AIP, atherogenic index of plasma; BMI, body mass index; CRP, C-Reactive Protein; GFR, glomerular filtration rate; Hb, hemoglobin; HDL-C, high-density lipoprotein cholesterol; IL, interleukin; LDL-C, low-density lipoprotein cholesterol; MCP-1, monocyte chemoattractant protein-1; POx, plasma oxalic acid; OSI, oxidative stress index; RAAS, renin–angiotensin–aldosterone system; SH group, sulphydryl group; TC, total cholesterol; tIS, total indoxyl sulfate.

**Table 2 medicina-59-02189-t002:** Univariate and multivariate Cox regression analysis of baseline markers for CVD event risk factors in the study group.

	Univariate Cox Regression Analysis			Multivariate Cox Regression Analysis		
Variable	Wald χ^2^	*p*-Value	HR (95% CI)	Wald χ^2^	*p*-Value	HR (95% CI)
Age, years	2.15	0.041	1.05 (1.02; 1.09)	1.11	0.290	1.01 (0.93; 1.26)
Systolic blood pressure, mm Hg	0.09	0.765	0.99 (0.91–1.08)			
Diastolic blood pressure, mm Hg	0.90	0.343	0.95 (0.86; 1.05)			
Male sex	1.88	0.171	0.37 (0.09; 1.54)			
Dialysis modality	1.29	0.257	2.30 (0.55; 9.6)			
Dialysis duration, months	3.58	0.048	1.11 (1.00; 1.27)	1.83	0.061	1.31 (0.95; 2.1)
Anuria	2.03	0.064	0.86 (0.17; 4.3)			
Obesity (BMI ≥ 30 kg/m^2^)	6.28	0.012	6.22 (1.21; 10.1)	5.21	0.018	1.63 (1.1; 2.4)
Glucose_blood	0.05	0.825	0.93 (0.48; 1.79)			
Ca, mmol/L	2.81	0.082	0.93 (0.49; 1.77)			
Triglycerides, mmol/L	5.82	0.015	1.39 (1.09; 2.36)	3.59	0.022	1.46 (1.1; 3.4)
HDL-C, mmol/L	3.05	0.048	0.06 (0.008; 0.9)	2.71	0.039	0.05 (0.01; 0.89)
MDA, mmol/L	3.41	0.044	1.27 (1.0; 8.8)	2.29	0.066	1.25 (0.97; 7.16)
Ceruloplasmin, g/L	1.86	0.172	0.91 (0.54; 1.3)			
tIS, μmol/L	4.26	0.018	2.15 (1.18; 3.22)	3.79	0.026	1.73 (1.06; 4.11)
POx (µmol/L)	5.61	0.001	4.66 (1.2; 16.5)	3.12	0.017	3.65 (1.3; 8.9)
IL-6, pg/mL	1.83	0.045	1.13 (1.01; 1.28)	1.56	0.068	2.43 (0.94; 6.2)
MCP-1, pg/mL	1.61	0.038	1.11 (1.07; 1.19)	1.37	0.172	1.05 (0.97; 1.13)
CRP, g/L	0.17	0.680	1.02 (0.95; 1.08)			
Total fecal ODA, %/0.01 g	5.51	0.001	9.41 (2.3; 19.4)	3.89	0.003	4.1 (1.4; 16.3)

Abbreviations: CRP, C-reactive protein; HDL-C, high-density lipoprotein cholesterol; IL, interleukin; MCP-1, monocyte chemoattractant protein-1; ODA, oxalate-degrading activity; POx, plasma oxalic acid; tIS, total indoxyl sulfate.

**Table 3 medicina-59-02189-t003:** Blood lipid profile, POx, tIS, oxidative stress, and pro-inflammatory markers in patients with kidney failure based on total fecal ODA status.

	Total Fecal ODA Status		*p*-Value
	Positive (≥1%/0.01 g of Feces) (*n* = 17)	Negative (≤−1%/0.01 g of Feces) (*n* = 15)	
Lipid profile parameters			
TC, mmol/L	5.3 (4.8–6.0)	5.8 (4.1–6.6)	0.89
Triglycerides, mmol/L	1.4 (102.0–1.7)	2.6 (1.6–3.1)	0.002
LDL-C, mmol/L	2.8 (2.4–3.5)	2.9 (1.8–3.8)	0.9
HDL-C, mmol/L	1.45 (1.3–1.8)	1.03 (0.9–1.2)	0.0005
AIP	2.8 (1.8–3.5)	3.1 (2.3–4.0)	0.007
Oxidative stress markers			
MDA, mmol/L	179.5 (112.2–282.0)	474.3 (250.1–522.4)	0.008
Ceruloplasmin, g/L	0.18 (0.12–0.25)	0.09 (0.07–0.14)	0.003
Transferrin, g/L	2.1 (1.5–2.7)	2.3 (1.8–2.6)	0.74
SH groups, mmol/L	1.65 ± 0.4	1.66 ± 0.3	0.95
OSI, CU	2.6 ± 1.5	5.2 ± 2.6	0.001
Proinflammatory markers, tIS and POx concentrations			
CRP, mg/L	4.9 (3.3–12.9)	12.5 (10.3–15.1)	0.03
IL-6, pg/mL	0.18 (0.1–1.4)	2.1 (0.1–6.6)	0.01
MCP-1, pg/mL	285.6 (234.4–315.6)	342.0 (330.0–418.1)	0.0003
tIS, μmol/L	23.5 (15.0–42.2)	63.8 (39.8–95.0)	0.0001
POx, µmol/L	19.4 (4.7–30.7)	49.5 (30.3–70.5)	<0.0001

The values are expressed as mean ± standard deviation (M ± SD) or as the median and interquartile range (Me (Q25–Q75)). The values are compared between the groups using Chi-square tests, Student’s *t*-test and the Mann–Whitney U test as appropriate. Abbreviations: AIP, atherogenic index of plasma; CRP, C-reactive protein; HDL-C, high-density lipoprotein cholesterol; IL, interleukin; LDL-C, low-density lipoprotein cholesterol; MCP-1, monocyte chemoattractant protein-1; ODA, oxalate-degrading activity; POx, plasma oxalic acid; OSI, oxidative stress index; SH group, sulphydryl group; TC, total cholesterol; tIS, total indoxyl sulfate.

## Data Availability

The current study dataset is available upon reasonable request to the corresponding author.

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
