# Peer review of "Gut Microbiota’s Oxalate-Degrading Activity and Its Implications on Cardiovascular Health in Patients with Kidney Failure: A Pilot Prospective Study"

_medicina, 2023, doi:10.3390/medicina59122189_

Round 1
Reviewer 1 Report
Comments and Suggestions for Authors
In this pilot prospective study, Natalia Stepanova et al. provide novel insights into the relationship between gut microbiota's oxalate degrading activity and cardiovascular health in patients undergoing kidney replacement therapy.
The figures are very clear and aesthetically pleasing. The flowchart in particular is very useful to the reader.
The article is very well written, with just a few grammatical errors. The main findings of their research are well emphasized in the discussion.
On page 8, values in the table are obscured with numbering.
The analysis itself is very detail-oriented and well thought out.
It is a shame that there were only 18 controls in the study. Nevertheless, I hope to see this study in the future with a large increase in participants.
Comments on the Quality of English LanguageA few oversights were spotted. Nothing major.
Author Response
Thank you for your constructive feedback on our study. Your positive remarks are invaluable, and we appreciate the time and effort you dedicated to evaluating our work. We also appreciate your keen observation regarding the numbering obscuring values in the table on page 8. However, the PDF file we uploaded did not have this omission, which probably occurred after repeated file conversions. In any case, we addressed this oversight in the final version of the manuscript.
We share your comment about the limited number of controls and agree that expanding the participant pool would strengthen the generalizability of our findings. In future studies, we plan to address this limitation by increasing the sample size, and we thank you for highlighting the importance of this aspect.
Thank you for your time and expertise.
Reviewer 2 Report
Comments and Suggestions for Authors
Dear authors,
I congratulate you on a very well designed pilot study on the effects of gut microbiota and cardiovascular complications in patients with CKD. As the authors mentioned there are no study on this topic in current literature and determining an appropriate sample size was an issue but, as in most similar cases, the authors overcame this by including a standard sample size. The methodology of this study as well as the results are well presented and the discussion is up to date.
Author Response
Thank you for your kind words on our study. We sincerely appreciate your positive feedback and acknowledgment of the effort we put into designing and executing the study.
Reviewer 3 Report
Comments and Suggestions for Authors
In the manuscript: “Gut microbiota's oxalate-degrading activity and its implications on cardiovascular health in patients with kidney failure: A pilot prospective study”. The authors describe the association between gut microbiota's oxalate degrading activity (ODA) and the risk of developing CVD over a 3-year follow-up period in a cohort of patients undergoing KRT.
It is an interesting topic that contributes to knowledge in the area, but I will make some recommendations to improve the quality of the manuscript.
The introduction gives a good summary of the role of dysbiosis in the intestinal microbiota and its relation to the development of cardiovascular complications. But I see the need in the paragraph between lines 68 to 82 to add a comment on the differences between the intestinal microbiota in CKD patients with and without renal replacement therapy.
The section on materials and methods is very well described, I have no recommendations. However, it strikes me that the mean blood flow rate is so low at 300 even though the inclusion criteria for the Kt/V target is at least 1.4 for hemodialysis patients, please add a comment on this.
The section of resulta ist correct.
In the discussion section, I recommend at the end of the discussion, to add a commentary on the therapeutic interventions that can be used to improve dysbiosis in renal replacement therapy patients (Prebiotics, Probiotics and Synbiotics).
I recommend reviewing this article:
Review Nefrologia 2017 Jan-Feb;37(1):9-19. doi: 10.1016/j.nefro.2016.05.008. Epub 2016 Aug 21. Gut microbiota in chronic kidney disease. Secundino Cigarran Guldris, Emilio González Parra, Aleix Cases Amenós. PMID: 27553986 DOI: 10.1016/j.nefro.2016.05.008
Comments on the Quality of English LanguageMinor editing of English language required
Author Response
In the manuscript: “Gut microbiota's oxalate-degrading activity and its implications on cardiovascular health in patients with kidney failure: A pilot prospective study”. The authors describe the association between gut microbiota's oxalate degrading activity (ODA) and the risk of developing CVD over a 3-year follow-up period in a cohort of patients undergoing KRT.
It is an interesting topic that contributes to knowledge in the area, but I will make some recommendations to improve the quality of the manuscript.
Thank you for your constructive feedback on our study and for the time and effort you dedicated to evaluating our work.
The introduction gives a good summary of the role of dysbiosis in the intestinal microbiota and its relation to the development of cardiovascular complications. But I see the need in the paragraph between lines 68 to 82 to add a comment on the differences between the intestinal microbiota in CKD patients with and without renal replacement therapy.
Thank you for your valuable recommendation. The suggested text has been incorporated into the Introduction section (L 71-90): “CKD is widely acknowledged to cause gut microbiota dysbiosis, which is evident in the early stages of CKD and progresses with structural and functional changes, culminating in distinct alterations during KRT [3,4,24,25]. Patients undergoing KRT exhibit a notable reduction in overall gut microbiota diversity, marked by a decline in anti-inflammatory butyrate-producing microbes (e.g., Roseburia, Prevotella, Bacteroides, Eubacterium) and an increase in pro-inflammatory microbes (Proteobacteria, Actinobacteria) [24,25]. These changes, coupled with commonly reported alterations such as decreased quantities of Lactobacillaceae and Bifidobacteriaceae, may impact the gut microbiota's ODA and result in elevated plasma oxalic acid (POx) levels [7,23]. Concurrently, both gut microbiota dysbiosis and hyperoxalemia have been shown to be associated with CVD and all-cause mortality in patients with kidney failure [26–28]. CKD-induced gut microbiota dysbiosis exacerbates toxin accumulation, compromises the integrity of the mucosal barrier, disrupts immune tolerance, and intensifies systemic inflammation, contributing to the development of CVD and increased mortality in patients with kidney failure [24,29,30]. In turn, hyperoxalemia is hypothesized to potentially lead to endothelial dysfunction, inflammation, fibrosis, and altered hemodynamic parameters, suggesting a plausible pathophysiological link between oxalate and impaired cardiac function [28,31,32]. These findings prompt the exploration of whether gut microbiota ODA could serve as a potential link between hyperoxalemia and CVD in patients with kidney failure.”
The section on materials and methods is very well described, I have no recommendations. However, it strikes me that the mean blood flow rate is so low at 300 even though the inclusion criteria for the Kt/V target is at least 1.4 for hemodialysis patients, please add a comment on this.
Thank you for pointing out this omission. In our center, the typical blood flow rate for most patients is between 5 and 7 ml/kg/min. In this study, we included patients with a minimum blood flow rate of 300 ml/min. This was a typographical error and we have now replaced "the median blood flow rate" with "the minimum blood flow rate."
The section of resulta ist correct.
Thank you for your positive assessment.
In the discussion section, I recommend at the end of the discussion, to add a commentary on the therapeutic interventions that can be used to improve dysbiosis in renal replacement therapy patients (Prebiotics, Probiotics and Synbiotics).
Thank you for your valuable recommendation. The following paragraph has been added to the Discussion section (L 471-481): “One possible avenue that arises from our findings for therapeutic intervention is the use of prebiotics, probiotics, and synbiotics to modulate the composition and function of the gut microbiota, potentially improving oxalate metabolism and reducing cardiovascular risk in this patient population. Several studies have demonstrated the efficacy of prebiotics, probiotics, and synbiotics in patients with kidney failure, showing promising results in modulating the composition of the gut microbiota, reducing uremic toxins and inflammation, and preserving residual renal function [29,62–65]. However, it is crucial to emphasize that the exact impact of these interventions on ox-alate metabolism [18, 23] and the mitigation of cardiovascular risk [65] is not fully understood. This field of knowledge remains contentious and is continually evolving, necessitating further research to identify specific strains, dosages, and treatment dura-tions [29, 62, 65].”
I recommend reviewing this article:
Review Nefrologia 2017 Jan-Feb;37(1):9-19. doi: 10.1016/j.nefro.2016.05.008. Epub 2016 Aug 21. Gut microbiota in chronic kidney disease. Secundino Cigarran Guldris, Emilio González Parra, Aleix Cases Amenós. PMID: 27553986 DOI: 10.1016/j.nefro.2016.05.008
Thank you for your suggestion. We have incorporated this valuable resource into our work, and it is indeed contributing to the enrichment of our research.
Reviewer 4 Report
Comments and Suggestions for Authors
This is a well-written article. The authors presented their results with the appropriate statistical analysis, while the discussion section covered the study's relevant references, strengths, and limitations.
Below, there are some minor points.
- During three years, no withdrawals?
- Is there any more specific information about their medication? Was the same for the study group during the three years?
- Lipid-lowering therapy, please specify.
- Please provide clinical and laboratory data after three years, not only at baseline.
Author Response
This is a well-written article. The authors presented their results with the appropriate statistical analysis, while the discussion section covered the study's relevant references, strengths, and limitations.
Thank you for your positive remarks and constructive comments. We appreciate the time and effort you dedicated to evaluating our work.
Below, there are some minor points.
- During three years, no withdrawals?
Thank you for your question. As it was an observational study with no additional treatment and considering the limited sample size from a single center, none of the patients withdrew from the study. All participants continued their treatment with the same dialysis modality throughout the follow-up period.
- Is there any more specific information about their medication? Was the same for the study group during the three years?
Thank you for your valuable suggestion and question. In response to your suggestion, we have enhanced the Medication section in Table 1, providing additional details on medications used at enrollment, and the relevant data have been included in the text (L 292-296).
Indeed, there were modifications in pharmacological prescriptions throughout the study period. Antihypertensive drugs, phosphate binders, erythropoietins, and iron supplementation were adjusted based on clinical needs. However, as outlined in the Methods section, none of the participants received medications that could influence ODA, such as antibiotics, probiotics, or immunosuppressive treatment.
- Lipid-lowering therapy, please specify.
Thank you for your suggestion. At the time of study enrollment, 11 patients had been prescribed atorvastatin or rosuvastatin before initiating dialysis, and their prescription was not restricted based on high cardiovascular risk. The use of statins was mentioned in the text of the first manuscript version. However, following your recommendation, we have updated "Lipid-lowering therapy" in Table 1 to explicitly indicate "Statins."
- Please provide clinical and laboratory data after three years, not only at baseline.
Thank you for your recommendation. Unfortunately, conducting a longitudinal study for all markers included in the research was not initially planned, and we did not measure them repeatedly after 3 years.